# Biosorbents from Plant Fibers of Hemp and Flax for Metal Removal: Comparison of Their Biosorption Properties

**DOI:** 10.3390/molecules26144199

**Published:** 2021-07-10

**Authors:** Chiara Mongioví, Nadia Morin-Crini, Dario Lacalamita, Corina Bradu, Marina Raschetti, Vincent Placet, Ana Rita Lado Ribeiro, Aleksandra Ivanovska, Mirjana Kostić, Grégorio Crini

**Affiliations:** 1Laboratoire Chrono-Environnement, Faculté des Sciences & Techniques, UMR 6249, Université Bourgogne Franche-Comté, 16 route de Gray, 25000 Besançon, France; chiara.mongiovi@univ-fcomte.fr (C.M.); nadia.crini@univ-fcomte.fr (N.M.-C.); dario.lacalamita@univ-fcomte.fr (D.L.); 2PROTMED Research Centre, Department of Systems Ecology and Sustainability, University of Bucharest, Spl. Independentei 91–95, 050095 Bucharest, Romania; corina.bradu@g.unibuc.ro; 3FEMTO-ST, CNRS/UFC/ENSMM/UTBM, Department of Applied Mechanics, Université Bourgogne Franche-Comté, 16 route de Gray, 25000 Besançon, France; marina.raschetti@univ-fcomte.fr (M.R.); vincent.placet@univ-fcomte.fr (V.P.); 4Laboratory of Separation and Reaction Engineering-Laboratory of Catalysis and Materials (LSRE-LCM), Faculdade de Engenharia, Universidade do Porto, Rua Dr. Roberto Frias s/n, 4200-465 Porto, Portugal; ritalado@fe.up.pt; 5Innovation Center of the Faculty of Technology and Metallurgy, University of Belgrade, Karnegijeva 4, 11000 Belgrade, Serbia; aivanovska@tmf.bg.ac.rs; 6Department of Textile Engineering, Faculty of Technology and Metallurgy, University of Belgrade, Karnegijeva 4, 11000 Belgrade, Serbia; kostic@tmf.bg.ac.rs

**Keywords:** lignocellulosic fiber, flax, hemp, felt, biosorption, metals

## Abstract

Lignocellulosic fibers extracted from plants are considered an interesting raw material for environmentally friendly products with multiple applications. This work investigated the feasibility of using hemp- and flax-based materials in the form of felts as biosorbents for the removal of metals present in aqueous solutions. Biosorption of Al, Cd, Co, Cu, Mn, Ni and Zn from a single solution by the two lignocellulosic-based felts was examined using a batch mode. The parameters studied were initial metal concentration, adsorbent dosage, contact time, and pH. In controlled conditions, the results showed that: (i) the flax-based felt had higher biosorption capacities with respect to the metals studied than the hemp-based felt; (ii) the highest removal efficiency was always obtained for Cu ions, and the following order of Cu > Cd > Zn > Ni > Co > Al > Mn was found for both examined biosorbents; (iii) the process was rapid and 10 min were sufficient to attain the equilibrium; (iv) the efficiency improved with the increase of the adsorbent dosage; and (v) the biosorption capacities were independent of pH between 4 and 6. Based on the obtained results, it can be considered that plant-based felts are new, efficient materials for metal removal.

## 1. Introduction

In Europe, the Water Framework Directive (WFD) of 2000 established guidelines for the protection of surface water, underground water and coastal water. The main objective was to maintain and/or to restore water quality in Europe. In France, the WFD has been incorporated into French legislation by several laws, the latest of which dates from July 27, 2015 and deals with chemical substances [1,2,3]. This regulation lists two main categories of unwanted substances: dangerous priority substances (DPS) considered persistent, highly toxic or causing bioaccumulation and priority substances (PS) presenting a significant risk for the environment. A total of 70 substances are concerned, of which 8 are metals, namely Hg and Cd as DPS; Pb and Ni as PS; and As, Cr, Cu and Zn. The surface treatment industry is a particular contributor to the release of these metals [2], and among the various industrial activities, this sector is still considered to be one of the largest consumers and polluters of water [1,2,3]. In addition, these eight metals are also the subject of particular attention insofar as they account for the calculation of the so-called metox index, an indicator used to quantify certain forms of toxic pollution and to calculate the taxes that certain high-risk industries must then pay to the French Water Authority. The World Health Organization (WHO) and other environmental protection agencies (US EPA) have also adopted guidelines on the release of metals and their presence in water compartments.

Currently, Europe is asking industrials to innovate by reducing and/or eliminating metals present in their discharge waters at trace levels. In theory, there are methods available including membrane filtration, evaporation, liquid-liquid extraction, adsorption and ion exchange [4,5,6,7,8,9]. For example, the surface treatment industries can upgrade their standard installations used to treat their wastewater before release by using the series of (sand) filtration + adsorption onto activated carbons + ion-exchange to tend toward zero pollution [6,8]. This technology is recognized for its effectiveness, but its widespread use is very limited due to its high cost. It is also extremely difficult, while being economically viable, to remove substances that are heterogeneous and variable in nature and that occur at very low concentrations in discharge waters. In fact, economic considerations must be examined because the industrial sector in Europe is characterized by a large proportion of small and medium-size enterprises that are not very enthusiastic about investing in additional equipment with high operating and maintenance costs. In recent years, many attempts have been made to find inexpensive alternative materials as biosorbents that are both effective and acceptable for industrial use from both a technological (simplicity) and economic point of view. Numerous reviews on this topic can be consulted [10,11,12,13,14,15,16,17,18,19].

A recent class of materials proposed for environmental applications such as the removal of pollutants from solutions is based on the use of plants such as hemp and flax. These two plants are annual, high-yielding crops grown for their fibers and seeds [20,21,22]. Hemp and flax are interesting raw, eco-friendly, lignocellulosic materials because of their ease of production (rapid growth, no pesticides), low cost, renewable character, particular chemical composition of their fibers (mainly cellulose, hemicelluloses and lignin), particular structure (multicellular fibers of fibrilar structure), physical and mechanical properties as well as their versatility. They are usable in the form of powder, fragments (shives), fibers and oils since the entire plant (seeds and plant stem) is recoverable [20]. Like other lignocellulosic fibers or plant fibers such as jute, ramie, sisal, kenaf and bamboo, hemp and flax contain a high content of cellulose and comprise three main constituents (cellulose, hemicelluloses, and lignin) and other minor components. Within the framework of a circular and ecological economy, products based on natural fibers of vegetal origin such as hemp and flax are attracting great interest for their numerous applications in various industrial fields such as textiles, the paper industry, the construction sector, composites (automotive applications, plastic, packaging), agrochemistry, mulch and animal bedding [20,21,22,23,24,25,26]. However, in the field of wastewater treatment, there are no real applications yet, but there is a growing interest in the academic field [25,26].

Indeed, hemp and flax in fiber form have been proposed for metallic ion removal from aqueous synthetic solutions [23,25,26,27,28,29,30,31,32,33,34,35]. Kostić’s group studied the capacity of hemp for removing Zn, Pb and Cd ions from aqueous monometallic solutions and indicated strong bonding of metal ions to functional groups such as the carboxylic, carbonyl and hydroxyl groups present in the hemp structure (cellulose, hemicelluloses, lignin and extractives) [27,32,35]. In a series of papers on the ability of hemp to act as a non-conventional, low-cost and effective adsorbent, Păduraru and co-workers also reported similar conclusions for other metals such as Cu, Cr, Ag and Co [28,29,30]. Balintova et al. [31] studied the removal of copper ions by hemp and reported biosorption capacities between 3.91 and 4.45 mg/g, which were similar to those of the conventional commercial materials used in the treatment of metal contaminated water. Other studies have reported similar findings and conclusions [23,24,33,34]. Flax has also been proposed for metal removal [36,37,38,39,40,41]. For example, Abbar et al. [36,37] showed that 1 g of flax was capable of adsorbing 8.4 mg of Zn, 9.9 mg of Cu and 10.7 mg of Pb. The fibers were brought into contact with the metals present in water at a pH between 4 and 7 (depending on the element to be complexed) for 1 h under agitation. The authors reported that the chemical groups present in the structure of polysaccharides were responsible for biosorption, similarly to the case of hemp fibers. Melia et al. [38] also showed that flax byproducts were interesting as unconventional materials for adsorbing Cd: For metal concentrations between 1.1 and 21.5 mg/L, flax was capable of removing more than 90% of Cd in less than 10 min for the lowest investigated concentrations. The maximum capacity reported by the authors was 3.36 mg/g. Abutaleb et al. [40] recently reported a maximum capacity of 40.9 mg/g for U adsorption onto flax fibers. All these results were obtained using hemp and flax in the form of fibers.

Recently, our group proposed for the first time the use of hemp-based biosorbents in a felt form to treat monometallic and polymetallic aqueous solutions [42,43,44,45,46]. Results of biosorption in batch mode showed that hemp-based felts could be used as efficient biosorbents for the removal of metals present in synthetic solutions [42,43] or in real effluents [44,45,46]. Pursuing our interest in such plant-based biosorbents, in this study, we compared the biosorption performance of two lignocellulosic-based felts (Figure 1) made of hemp and flax fibers and a smaller amount of synthetic fibers toward the removal of 7 metals, namely Al(III), Cd(II), Co(II), Cu(II), Mn(II), Ni(II) and Zn(II). These metals were selected based on their common presence in discharge waters from the French metal industry. Moreover, Cd, Cu, Ni and Zn are also on the list of priority pollutants defined by the French Water Agency [2]. Herein, studies concerning the effects of metal concentration, adsorbent dose, contact time and pH were evaluated using the batch method. To our knowledge, this is the first report on the comparison of metal removal from solutions by hemp- and flax-based materials in felt form, which might be an interesting option to apply in industrial scenarios due to its easy operation as a filter.

## 2. Materials and Methods

### 2.1. Materials and Chemicals

Hemp- and flax-based felts were provided by a French lignocellulosic material processing company in Franche-Comté (Eurochanvre, Arc-les-Gray, Haute-Saône) (Figure 1). Before use, the two felts were extensively washed in water (pH = 5.8 ± 0.1) and then dried at 60 °C until constant weight [42]. Their characteristics are reported in Table 1.

The chemical composition of felts was determined according to a procedure given in the literature [43] by subsequent removal of non-cellulosic components and evaluation of the weight loss: water solubles (extraction with boiling water for 30 min), fats and waxes (extraction with dichloromethane for 4 h), pectins (extraction with 1% ammonium oxalate at boiling temperature for 1 h), lignin (extraction with 0.7% sodium chlorite (pH 4.0–4.5) at boiling temperature for 2 h) and hemicelluloses (treatment with 17.5% sodium hydroxide at room temperature for 45 min). After removal of the non-cellulosic components, α-cellulose remained as a solid residue. For each sample, the chemical composition was determined in duplicate.

The ion exchange capacity (IEC in Table 1) of samples was determined by a potentiometric titration previously described in the literature [42].

Elemental analysis of the sample surfaces was performed using the Thermo NORAN system for energy-dispersive X-ray spectroscopy and electron beam excitation with a voltage from 15 keV to 20 keV. The surfaces of samples were examined on a scanning electron microscope (Apreo, ThermoFisher Scientific) with a tungsten filament voltage from 15 keV to 20 keV and low-vacuum conditions.

Metal sulfate salts were purchased from Sigma-Aldrich France and used as received. Appropriate weights of each metal were dissolved in water to obtain a stock solution containing 300 mg/L. Solutions having lower metal concentrations (range 1–100 mg/L) were obtained by dilution of the stock solution. The metal concentration in all replicates of each experiment was analyzed by ICP-AES following a standard protocol prior to each experiment [42,43,44,45,46].

### 2.2. Biosorption of Metals by Batch Technique

Metal removal from solutions by biosorption on the two felts was studied using a batch method [42,43]. In each experiment, a fixed amount of felt was introduced into 100 mL of an aqueous solution containing a known concentration of each metal. The mixture was then mechanically stirred for a given time on a rotating shaker at a constant speed (250 rpm) and at room temperature (22 ± 1 °C). After treatment, the felt was easily removed, and the metal concentration within the solution was determined. The biosorption performance was then calculated and expressed as removal/reduction in a percentage (as a ratio between the amount of metal adsorbed and the amount of metal initially present in the solution). In controlled conditions, the initial metal concentration (1, 2, 5, 10, 15, 20, 25, 35, 50, 65, 75 and 100 mg/L), adsorbent dose (0.25, 0.50, 1.00, 1.50 and 2.00 g), contact time (range 1–60 min) and pH (4, 5 and 6) in the solution were varied to investigate their effect on biosorption capacity. The concentrations and pH values studied are representative of the values found in industrial effluents from the surface treatment industry. All experiments were replicated (*n* = 3–5) under identical conditions (*n* in the figure captions corresponds to the number of experiments). At the end of the biosorption treatment, a slight pH variation of between 0.4 and 0.7 units (single solutions) was observed in each experiment.

### 2.3. Biosorption Equilibirum

The two most common types of isotherm models used in the literature are the Langmuir and the Freundlich. Their non-linear equations are given below: $$\mathrm{Langmuir}:\;q_{e}=\frac{x}{m}=\frac{K_{L}C_{e}}{1+a_{L}C_{e}}$$
$$\mathrm{Freundlich}:\;q_{e}=K_{F}C_{e}{}^{1/n_{F}}$$
where *x* is the amount of metal adsorbed (mg), *m* is the amount of material used (g), *C_e_* (mg/L) and *q_e_* (mg/g) are the liquid phase concentration and solid phase concentration of adsorbate at equilibrium, respectively, *K_L_* (L/g) and *a_L_* (L/mg) are the Langmuir isotherm constants, *K_F_* is the Freundlich constant (L/g) and 1/*n_F_* is the heterogeneity factor.

According to the Langmuir model, the adsorption process takes place at specific homogeneous sites on the adsorbent surface until a complete monolayer is formed. This model is used to estimate the maximum adsorption capacity *q_max_* (mg/g), which corresponds to the adsorbent saturation, and it is numerically equal to *K_L_/a_L_*. The Freundlich model can be used to describe adsorption on heterogeneous surfaces, and it is not restricted to the formation of a monolayer. The *n_F_* value indicates the degree of non-linearity between the solution concentration and biosorption; in particular, if *n_F_* = 1, the biosorption is linear; if *n_F_* < 1, the biosorption process is chemical, whereas, if *n_F_* > 1, biosorption is a favorable physical process.

## 3. Results and Discussion

### 3.1. Characterization of Hemp- and Flax-Based Felts

Before the biosorption experiments, the hemp- and flax-based felts were characterized in terms of their structural properties; chemical composition; contents of C, N and S; and ion exchange capacity. From the results listed in Table 1, it is evident that hemp-based felt was thicker and had a higher surface weight than flax-based felt. Both two studied felts contained lignocellulosic fibers (hemp or flax) as well as a certain amount of synthetic fibers (Table 1). Hemp-based felt was characterized by a lower content of synthetic fibers, whereby the hemp fibers had a lower content of α-celluloses and consequently, a higher content of non-cellulosic components (such as hemicelluloses, lignin, pectins, fats and waxes as well as water solubles) compared to the flax fibers. Moreover, both felts had almost similar contents of C, N and S. The results obtained for ion exchange capacity revealed that the hemp-based felt had significantly lower ion exchange capacity in comparison to the flax-based felt.

### 3.2. Comparison between Hemp- and Flax-Based Biosorbents Regarding Metal Removal

Figure 2 compares the removal of 7 metals present in single-component aqueous solutions by the two felts at an initially spiked metal concentration of 25 mg/L. The values of removal are expressed as percentage reduction. The data clearly shows that flax-based felt was a more efficient biosorbent than hemp-based felt, with values between 10 and 30% higher, depending on the metal. For both felts, Cu and Cd showed the highest affinity, whereas Mn and Al demonstrated the lowest level of removal. The reduction values for Cu and Cd were, respectively, 77 and 52% for hemp-based felt and 79 and 85% for flax-based felt. For the concentration studied, the same order of reduction was obtained for the two felts: Cu > Cd > Zn ~ Ni ~ Co > Al ~ Mn. Each experiment was repeated five times under identical conditions to guarantee the reproducibility of the experimental data.

In previous works [42,44], we have mainly considered a physisorption (surface biosorption and/or diffusion) mechanism to explain the performances of hemp felts. The difference in the degree of biosorption can be attributed to the physical and chemical characteristics of each metal such as ionic radii, molar mass and Allred–Rochow electronegativities, as shown in the literature data [24,28,32,33]. In addition, for copper, another interaction can occur. Indeed, in all experiments, a change in the pH of the solution was observed after contact with the two felts, favoring Cu biosorption. For Cu, the initial pH of 5 was raised to 5.6 for both felts. At these pH values, Cu can start to precipitate, explaining the higher reduction. The higher performance obtained for flax-based felt can also be explained by other interactions. Indeed, the carboxylate groups present in hemp- and flax-based felts can participate in the biosorption process through electrostatic interactions and ion-exchange. The 8.7 times higher ion exchange capacity of flax-based felt compared to that of hemp-based felt (IEC = 0.87 meq/g vs. 0.10 meq/g, Table 1) may explain the higher performance of flax compared to hemp-based felt. Both mechanisms (physisorption and chemisorption) are also directly dependent on the exchange surface. This is defined by the external surface of the felts, including a bundle of fibers. Within these non-woven fabrics and for a given mass, the exchange area directly depends upon the level of fiber individualization and separation as well as the fiber roughness. Figure 1 shows that the fineness of the flax fibers was significantly higher than those of hemp fibers. In addition to the fineness of the fibers, the influence of their molecular and microstructure on metal removal should be considered. Namely, flax fibers are characterized by a higher content of α-cellulose (Table 1), whose hydroxyl groups can form ion–dipole interactions with heavy metal ions [45]. Additionally, the effects of the hemp and flax microstructures, especially their lumen size and shape, on metal biosorption should not be neglected. Namely, the hemp fiber lumen is wider but cleft-shaped (i.e., closed), while the flax fiber lumen is smaller but round [46,47]. An analysis of these data showed that the mechanisms were complicated because they implicated the presence of different interactions (physical adsorption, electrostatic interactions, ion exchange, precipitation), and it was also quite possible that at least some of these interactions acted simultaneously to varying degrees.

### 3.3. Effect of Initial Metal Concentration

As discussed in the previous section, the biosorption of heavy metal ions by various lignocellulosic materials is dependent on the biosorbent itself. However, the experimental conditions (such as metal concentration, adsorbent dose, contact time and solution pH) play an important role in the overall biosorption process. Figure 3 shows the influence of initial metal concentration, in the range 1–100 mg/L, on the removal by the two felts of 7 metals in mono-contaminated aqueous solutions. The results clearly demonstrated that flax-based felt was more efficient as a biosorbent than hemp-based felt independently of the metal concentrations. For the concentration range studied, Cu showed the highest affinity, between 93 and 42% for hemp and between 100 and 69% for flax-based felt. At any concentration, the same order of removal was obtained for all metals studied. As expected, by increasing the metal concentration, biosorption efficiency decreased due to the saturation of the biosorption sites on the material. For example, when hemp-based felt was used and the initial metal concentration was increased from 10 up to 100 mg/L, the removal of Cd and Cu decreased from 74 down to 9% and 85 down to 42%, respectively. Considering the whole range of the concentrations studied, Cu exhibited stronger interactions for high concentrations compared to the other metals, whichever the felt used. Such results indicated the high affinity of the hemp- and flax-based felts for this metal, which was in agreement with our previous results [48,49,50]. Recently, Morin-Crini and co-workers [50] reviewed numerous studies indicating the better performance of certain natural biosorbents, such as plant-based materials containing polysaccharides, for Cu compared to other metals, according to the Irving–Williams series [24]. This empirical theory determines a series of stability constants of organo-metal complexes, pointing out the strong complexing of Cu by polysaccharides.

### 3.4. Effect of Adsorbent Dose

Figure 4 compares the removal of metals by the two felts by changing the dose of adsorbent in 100 mL of metal solution and keeping the other parameters constant. By increasing the amount of felt from 0.25 to 2 g, the removal percentage of each metal also increased, with superior performances for flax-based felt. One gram of a biosorbent was sufficient to obtain satisfactory biosorption results. The order of affinity obtained was always the same for all doses and each felt. Again, it is interesting to note the high levels of reduction obtained for Cu independently of the used felt. The flax-based felt also had a strong affinity for Cd complexing.

### 3.5. Effect of Contact Time

The biosorption data of metals versus contact time are presented in Figure 5, showing the time-dependent removal behavior in relation to the type of felt used for a concentration of 25 mg/L of each metal. Similar trends were obtained for hemp- and flax-based felts, indicating that both biosorption processes were rapid and uniform. Indeed, 10 min were sufficient to obtain biosorption equilibrium. This fast kinetic indicates a rapid binding of metal ions onto the felt, which suggests a surface biosorption mechanism. Similar results have been reported in the literature concerning the effect of contact time on the biosorption of single-component metal solutions by hemp and flax biosorbents in the form of fibers [24,27,28,33].

### 3.6. Effect of pH

Among other studied experimental conditions (metal concentration, adsorbent dosage and contact time), solution pH represented another important factor influencing biosorption. Namely, it affected the solubility of metal ions as well as the ionization state of the chemical groups present on the surface of the biosorbent [45]. In the current study, the effect of solution pH on metal reduction was studied at pH 4, 5 and 6, keeping the other parameters constant. This pH range was chosen because it corresponded to the pH values typically found in effluents from the surface treatment industry. The results reported in Figure 6 indicate that the performances of the two felts were almost independent of pH between 4 and 6. A similar observation was reported before for biosorption of Cd, Co and Zn onto hemp fibers [28,30,31] and biosorption of Cu and Zn onto flax fibers [36]. In each experiment, a change in the solution pH was systematically observed after contact with the felts, also suggesting a chemisorption mechanism [48,49,50,51,52]. The pH increase at the end of the experiments was less than 0.4 and less than 0.7 units for hemp- and flax-based felts, respectively.

### 3.7. Adsorption Isotherms

Experimental equilibrium data were fitted to the well-known and widely applied isotherm models of Langmuir and Freundlich. The isotherm parameters for the adsorption of metals onto materials are listed in Table 2. As expected, the Langmuir model was found to represent the equilibrium data with a much better fit as compared to the Freundlich model (Figure 7).

The R^2^ and χ^2^ values given in Table 2 confirmed that for both felts, the Langmuir model better fit experimental data. Based on the Langmuir analysis, for each metal, flax-based felt showed higher q_max_ values than those of hemp. The data demonstrated that both materials exhibited interesting biosorption properties toward Cu, which was the metal with the highest maximum adsorption capacity (4.51 mg/g and 5.53 mg/g for hemp- and flax-based materials, respectively). The Freundlich *n_F_* values showed that the adsorption process of metals onto the two felts was favorable under the studied conditions.

### 3.8. Tests with Polycontaminated Solutions

The potential of using hemp- and flax-based felts for removing the metals from an aqueous mixture of 7 metals was also studied. The experiments were conducted at two concentrations, 25 and 50 mg/L, corresponding to a total of 17.5 and 35 mg of metals in 100 mL of volume (Figure 8). The results showed that both felts (1 g in 100 mL) were even effective to treat a polycontaminated mixture. The performance values obtained for each metal in this solution were lower than those obtained in each corresponding single solution, indicating competition between the metals for complexing sites. However, the order of affinity did not change, with Cu always being the most retained and Mn and Al ions the least retained. At a total concentration of 175 mg/L (25 mg/L of each metal), flax-based felt had an excellent biosorption capacity, especially for Cu, with a reduction of 80% (value identical to that obtained in mono-contamination), and for Cd (65% lower concentration), even in the presence of other metals. When the total concentration was doubled to 350 mg/L, the reduction values decreased sharply, with the exception of those for Cu. Interestingly, when using 1 g of flax-based felt dispersed in 100 mL of a solution containing a total of 35 mg of metals, more than 70% of the 5 mg of copper present in the mixture was adsorbed. Similarly high values were published in the literature [53,54,55,56].

In both experiments, a change in the pH of the solution was also observed after contact with the felts, with the initial pH of 4.5 rising to 4.8 and 5.3 in the case of hemp- and flax-based felts, respectively. At these pH values, metals can start to precipitate, and this can participate in the biosorption mechanism.

In a recent paper, the interactions involved in the biosorption process using monocontaminated solutions were studied by employing scanning electron microscopy (SEM) and energy-dispersive X-ray spectroscopy (EDX), which revealed that both surface interactions and precipitation participate in metal removal from solutions [48]. Figure 9 compares the EDX spectra and SEM images of hemp- and flax-based felts before and after adsorption of a polycontaminated solution containing 7 metals. These data confirmed the adsorption of metals by the two materials.

## 4. Conclusions

This contribution reports a study of the use of hemp- and flax-based felts as biosorbents for metal removal. Under controlled experimental conditions, flax-based felt had higher biosorption capacities for the target metals (Al, Cd, Co, Cu, Mn, Ni and Zn) than the hemp-based felt. Indeed, in the case of flax-based felt, metal chemisorption occurred simultaneously with physical biosorption. A higher exchange surface area was also provided in the flax-based felt, characterized by higher fiber fineness and α-cellulose content when compared to hemp. In all experiments, Cu ions showed the highest removal efficiency for both biosorbents. The biosorption process was rapid because 10 min were sufficient to attain equilibrium. The biosorption capacities were also pH-independent in a range between 4 and 6. Based on these results, it can be considered that plant-based felts are new, efficient lignocellulosic materials for metal removal from aqueous solutions. More experiments will be carried out using other mono- or multi-component synthetic solutions and real effluents from the surface treatment industry. Here, the felts were used in batch processes, but our objective is to install the filters directly into an existing treatment tank under passive mode without additional energy consumption or process investments. In this case, our idea is not to regenerate the material but to eliminate it after use. Like all agricultural materials, after biosorption, felts can be incinerated to a much smaller volume of ash and/or recovery of adsorbed metal without adverse impact on the environment.

## Figures and Tables

**Figure 1 molecules-26-04199-f001:**
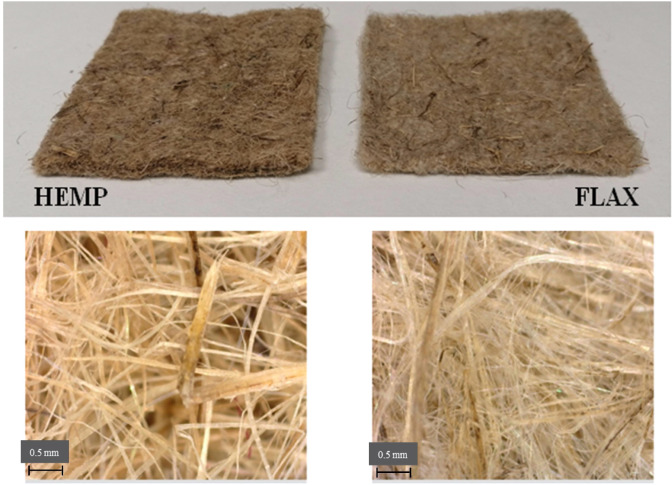
Hemp- and flax-based felts (below: optical microscopy images of the felt surface).

**Figure 2 molecules-26-04199-f002:**
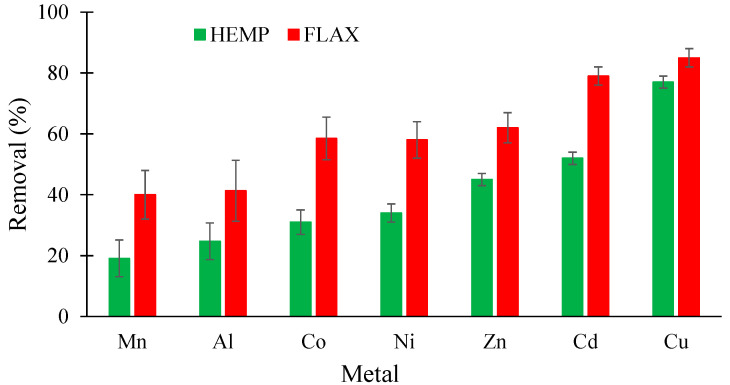
Comparison between removal (in %) of the seven metals by hemp and flax felts from single solutions using an initial metal concentration of 25 mg/L (other conditions: 1 g of felt in 100 mL of solution; contact time, 60 min; agitation speed, 250 rpm; temperature, 22 ± 1 °C; *n* = 5).

**Figure 3 molecules-26-04199-f003:**
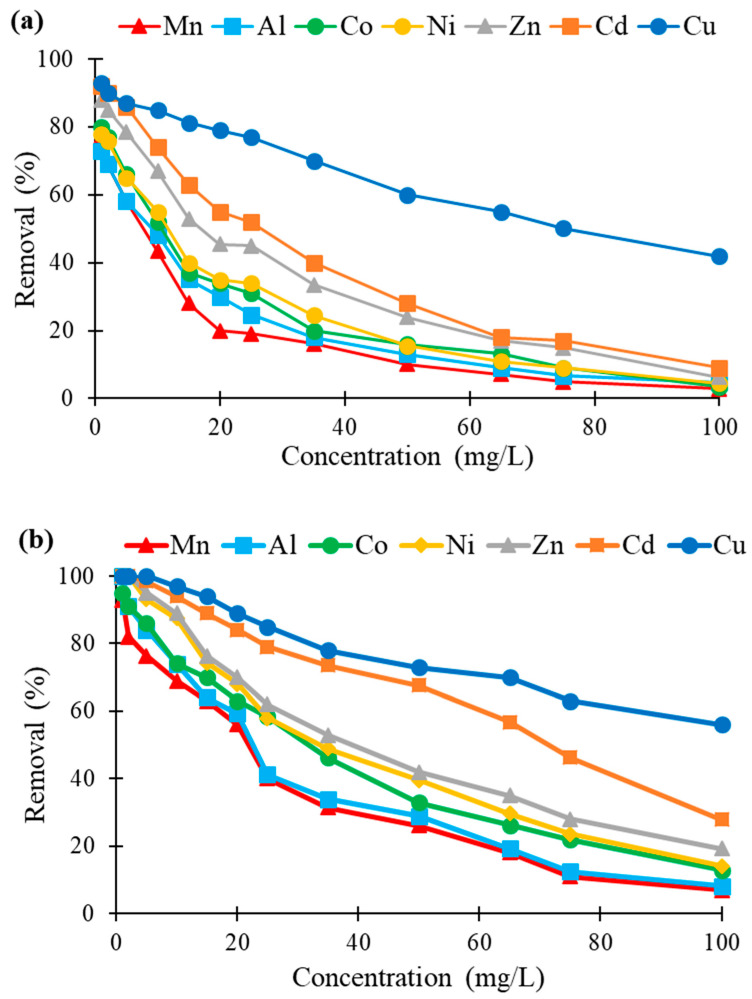
Effect of initial concentration (in mg/L) of seven metals (expressed in %) on their removal by (**a**) hemp- and (**b**) flax-based felts (other conditions: 1 g of felt in 100 mL of solution; contact time, 60 min; agitation speed, 250 rpm; temperature, 22 ± 1 °C; *n* = 3).

**Figure 4 molecules-26-04199-f004:**
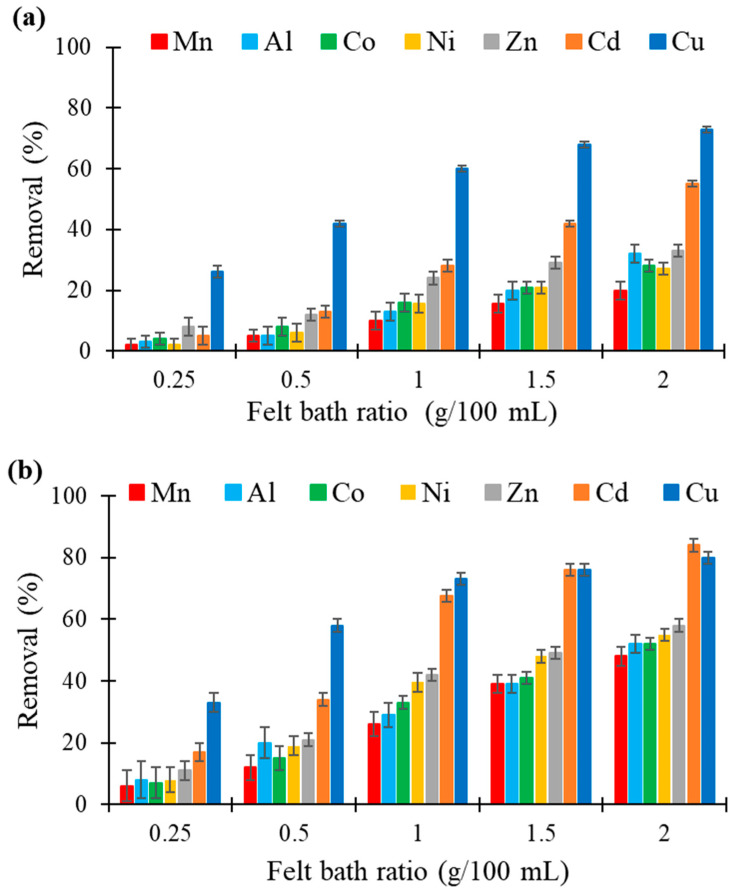
Comparison of the metal removal (in %) by different doses of (**a**) hemp- and (**b**) flax-based felts in 100 mL of mono-contaminated solutions with an initial metal concentration of 25 mg/L (other conditions: contact time, 60 min; agitation speed, 250 rpm; temperature, 22 ± 1 °C; *n* = 3).

**Figure 5 molecules-26-04199-f005:**
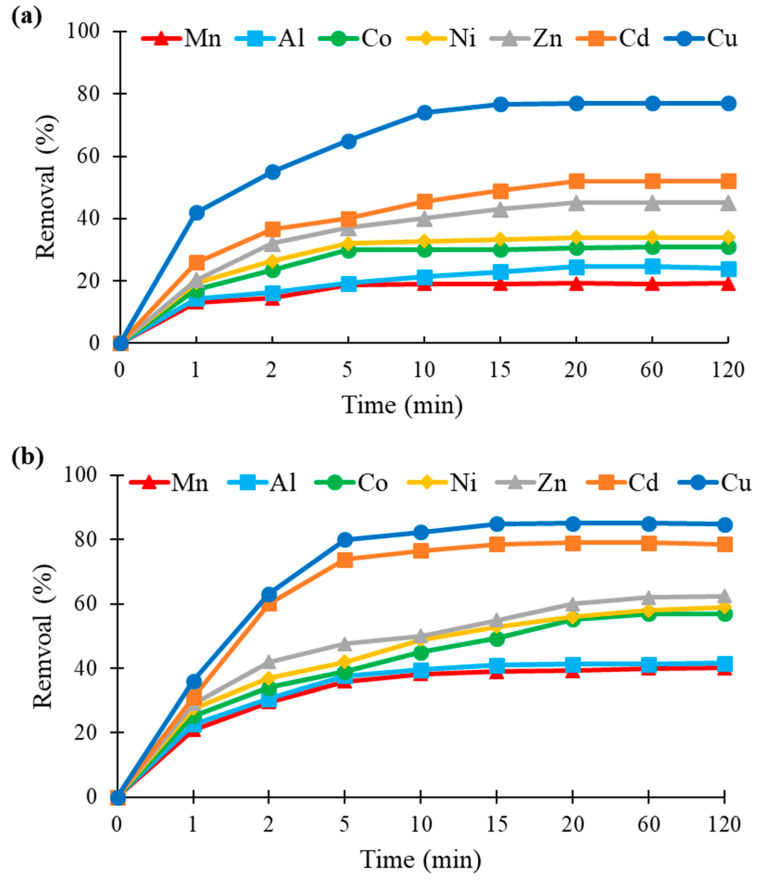
Kinetics of metal removal (in %) by (**a**) hemp- and (**b**) flax-based felts at an initial concentration of 25 mg/L (other conditions: 1 g of felt in 100 mL of solution; agitation speed, 250 rpm; temperature, 22 ± 1 °C; *n* = 3).

**Figure 6 molecules-26-04199-f006:**
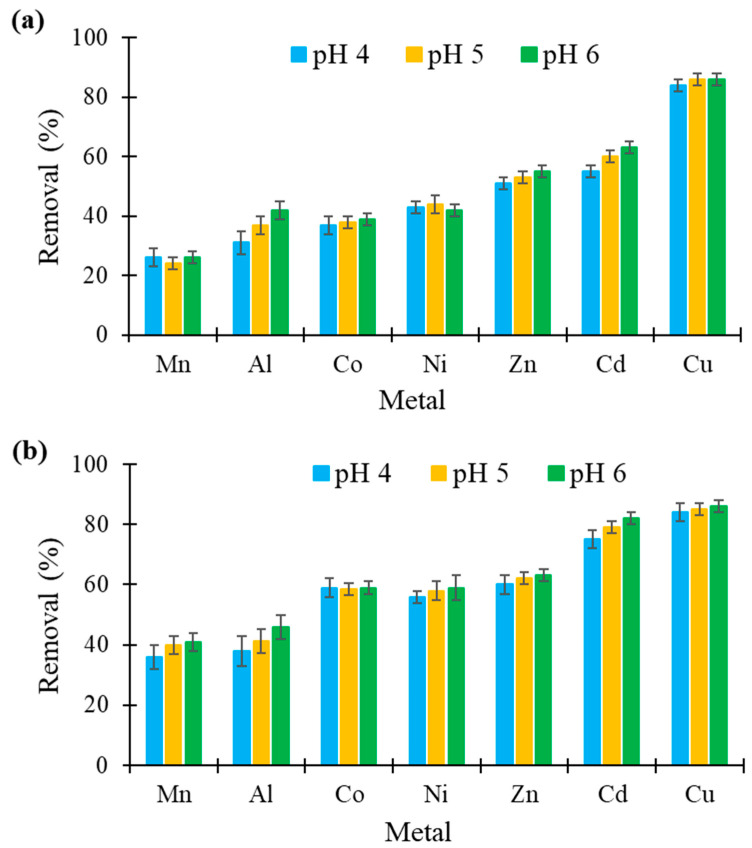
Influence of pH on the removal (in %) of seven metals by (**a**) hemp- and (**b**) flax-based felts from mono-contaminated solutions at an initial concentration of 25 mg/L (other conditions: 1 g of felt in 100 mL of solution; contact time, 60 min; agitation speed, 250 rpm; temperature, 22 ± 1 °C; *n* = 3).

**Figure 7 molecules-26-04199-f007:**
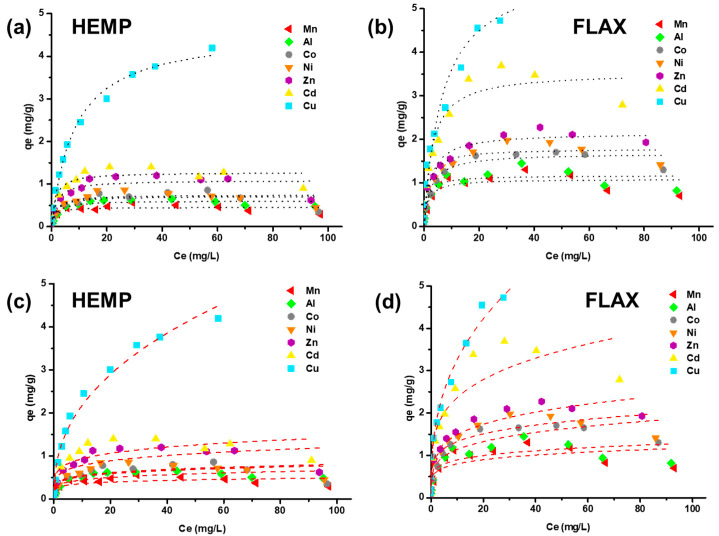
Comparison of Langmuir (**a**,**b**) and Freundlich (**c**,**d**) isotherms for 7 metals’ (Mn, Al, Co, Ni, Zn, Cd, Cu) adsorption onto hemp- and flax-based felts.

**Figure 8 molecules-26-04199-f008:**
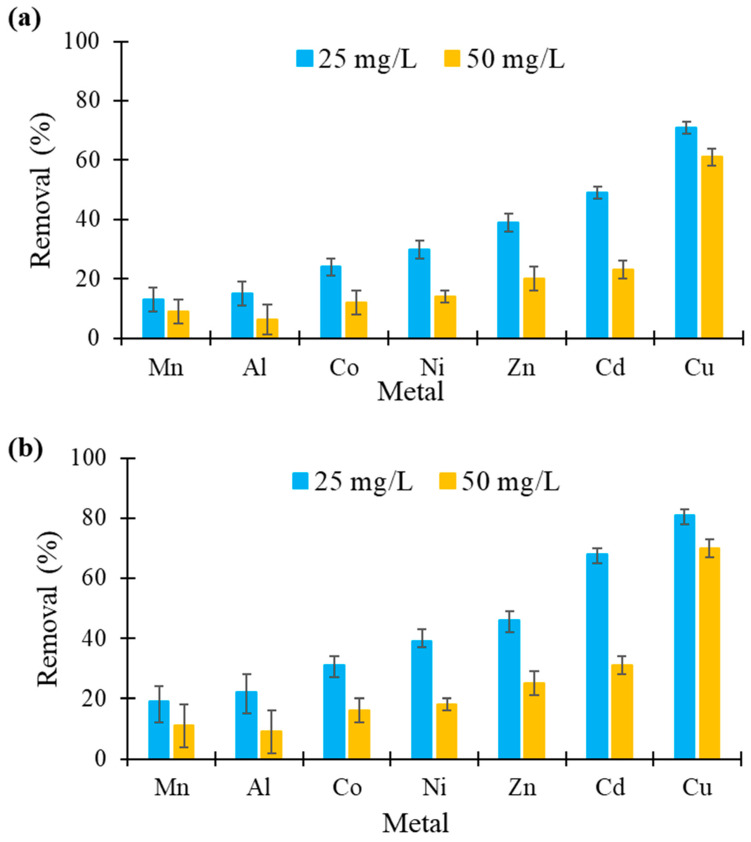
Comparison of metal removal (in %) from a solution containing a mixture of seven metals at two concentrations by (**a**) hemp- and (**b**) flax-based felts (other conditions: 1 g of felt in 100 mL of solution; initial concentration of each metal in solution, 25 or 50 mg/L; contact time, 60 min; agitation speed, 250 rpm; temperature, 22 ± 1 °C; *n* = 3).

**Figure 9 molecules-26-04199-f009:**
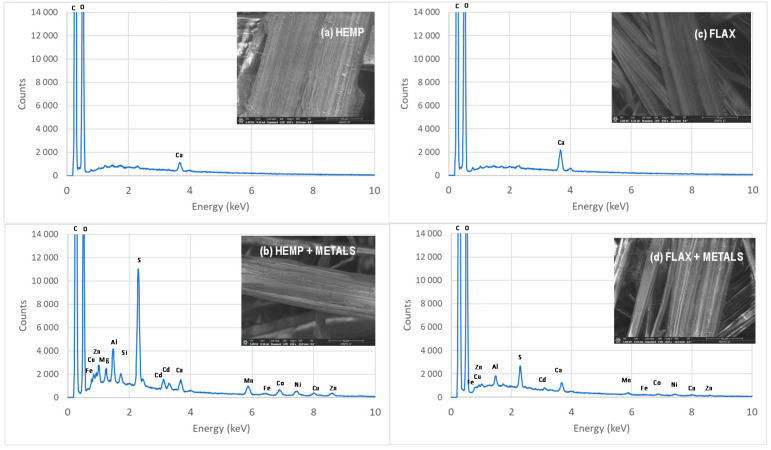
Elemental analysis using energy-dispersive X-ray spectroscopy and scanning electron microscopy images of hemp- and flax-based felts before (**a**,**c**) and after (**b**,**d**) adsorption of a solution containing a mixture of seven metals.

**Table 1 molecules-26-04199-t001:** Characteristics of hemp- and flax-based felts (washed samples).

	HEMP	FLAX
Thickness (mm)	~5	~3
Surface weight (g/m^2^)	665	280
Synthetic fiber content (%)	14.68	23.29
Lignocellulosic fiber content (%)	85.32	76.71
α-Cellulose * (%)	67.02 ± 0.2	72.77 ± 0.72
Hemicelluloses * (%)	19.32 ± 1.46	13.71 ± 1.00
Lignin * (%)	5.95 ± 0.33	6.93 ± 0.90
Pectins * (%)	1.50 ± 0.27	0.70 ± 0.31
Fats and waxes * (%)	1.23 ± 1.17	0.76 ± 0.12
Water solubles (%)	4.98 ± 0.18	5.12 ± 1.29
% C	41.1	43.5
% N	0.28	0.35
% S	0.06	0.09
Ion exchange capacity (meq/g)	0.10 ± 0.09	0.87 ± 0.09

* Chemical composition of lignocellulosic parts within the felts.

**Table 2 molecules-26-04199-t002:** Summary of the Langmuir and Freundlich isotherm constants and comparison of linear regression coefficients of determination (R^2^) and chi-square test statistics (χ^2^).

	Metal Ion	Langmuir					Freundlich			
		*q_max_* (mg/g)	*K_L_* (L/g)	*a_L_* (L/mg)	χ^2^	R^2^	*K_F_* (L/g)	*n_F_*	χ^2^	R^2^
HEMP	Mn	0.33	0.08	0.24	0.00567	0.9337	0.18	4.06	0.01258	0.6537
	Al	0.51	0.38	0.75	0.00398	0.9806	0.19	3.13	0.01554	0.7665
	Co	0.45	0.08	0.19	0.01860	0.8331	0.22	3.24	0.03517	0.7029
	Ni	0.53	0.15	0.28	0.01456	0.9207	0.22	3.10	0.03519	0.7284
	Zn	0.76	0.21	0.28	0.02939	0.8869	0.31	2.88	0.06693	0.7905
	Cd	1.02	0.57	0.56	0.02435	0.9563	0.38	2.88	0.07575	0.8237
	Cu	4.51	0.66	0.15	0.01485	0.9894	0.55	1.73	0.04286	0.9743
FLAX	Mn	0.78	0.27	0.35	0.03427	0.9416	0.33	3.03	0.07986	0.7684
	Al	0.90	0.38	0.42	0.03225	0.9564	0.45	3.82	0.07136	0.8545
	Co	1.45	13.24	9.13	0.01866	0.9738	0.43	2.6	0.07671	0.9020
	Ni	1.56	2.01	1.29	0.03575	0.9717	0.68	3.63	0.06871	0.9557
	Zn	2.05	2.09	1.02	0.03297	0.9914	0.72	3.33	0.04097	0.9764
	Cd	3.00	9.15	3.04	0.12359	0.9796	1.08	3.00	0.23426	0.9785
	Cu	5.53	1.75	0.32	0.02546	0.9906	1.40	2.70	0.18903	0.9802

## Data Availability

The data presented in this study are available on request from the corresponding author.

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
