# Peer review of "Biosorbents from Plant Fibers of Hemp and Flax for Metal Removal: Comparison of Their Biosorption Properties"

_molecules, 2021, doi:10.3390/molecules26144199_

Round 1

Reviewer 1 Report

This manuscript (molecules-1272787) presents the preparation of hemp- and flax-based felt materials as biosorbents for the removal of metals in aqueous solutions. The effect of initial metal concentration, adsorbent dosage, contact time and pH on the biosorption of Al, Cd, Co, Cu, Mn, Ni and Zn by the two lignocellulosic-based felts were investigated in detail. The preparation of lignocellulosic fibers extracted from plants are novel, and the flax-based felt had higher biosorption capacities with respect to the metals, especially Cu ion. However, there are still some issues to be addressed. Therefore, a major revision is suggested before its acceptance.

  1. Biosorption mechanism oflignocellulosic-based felts should be analyzed, and  its adsorption fits the Langmuir isotherm adsorption model or pseudo second order adsorption kinetic model? 
  2. The surface modification of lignocellulosic-based felts for bioadsorption should be considered.
  3. The effect of microstructure and properties of lignocellulosic-based felts on the biosorption capacities should be investigated.
  4. The reason of high biosorption capacities for Cu ion should be explored.
  5. The cycle performance of biosorptionof lignocellulosic-based felts should be investigated.
  6. The English (grammar and syntax) should be improved carefully.

Reviewer 2 Report

Dear Authors,

I would like to thank you for your submission to MDPI “molecules” journal. The article is generally well written and it was a pleasure to read it. The article examines the adsorption capacities of two felts from different biological materials for different metal ions with regard to water purification processes. It was shown that flax felts have a very good capability of reducing the copper ion concentration in aqueous solution and also certain affinities to other metal ions, such as Cd, Zn or nickel and minor ones with respect to Al or Mn. The authors also studied the effect of metal ion felt ratio concentration, pH value and the adsorption kinetics. It was shown that flax has superior adsorption properties compared to hemp felts, and in addition, the authors gave indications about the adsorption mechanisms and correlations between the felt composition to its performance. However, also some questions and comments did arise during a thorough examination and I would like to kindly ask the authors to reply on them.

  1. It would be easier for the reader to understand and interpret the results of the experiments, when the authors could state first the “Materials and Methods” section and then present the “Results and Discussion” section. Furthermore, it was not completely clear to me what materials were used for the adsorption studies? The felts were used to perform several dissolutions. I suppose the weight of the felt was measured before and after each treatment to calculate the content of cellulose, lignin, pectins, etc.? Then, did the authors us the original (untreated) felts for the adsorption studies or the material after the removal of all the other components? This should be stated very clearly. Finally, could the authors present a more detailed explanation how the ICP-AES was used to detect the ion concentration in the solutions, what repetitions were done (one sample measured several times or one adsorption experiment repeated 3-5 times and each sample analysed once) or what is the maximum precision of this method??
  2. The authors give a good and thorough overview about the studies done before in other work, but could the authors clearly state which aspects and hypothesis are not studied yet and are going to be examined in their study? In the last sentence of the Introduction, the authors state that it is the first study with felts, but why are they then interesting and what supporting characteristics would the authors expect from felts in the metal ion adsorption? Also, the images in Figure 1 need an information of the magnification in the microscope image or a scale. finally, in line 121, the authors also state that a lower amount of synthetic fibres were used- what are these (composition) and the ratio of hemp/flax fibres to synthetic ones?
  3. The presentation of Figure 3,4 and7 could be adopted in mg(metal ion) per gram of felt. This would clarify, for instance for Figure 3, that for the 1 g of felt used for these experiments the maximum amount of metal ion that can be adsorbed is reached, when a certain initial concentration of metal ions in solution is reached. From the current presentation, the reader cannot draw conclusions easily. Independently on the way to present the data, could the authors add grids to the y-axis to make the graphs easier readable, please?
  4. In case of Figure 4, it would be interesting to analyse how much grams of the felts are required to remove more than (for instance) 90% of the metal ions which adsorb well (Cu, Cd, Zn and Ni). This would also be a very interesting information in regard to its practical application
  5. Throughout the article, the authors mention several times that some metal species adsorb either via complex bonding, chemisorption or physisorption. Although a detailed examination of the adsorption mechanisms is out of the scope of this article, an interesting information, and simultaneously test for the strength of metal binding, would be to use 1 g of felt for several consecutive adsorption. This means that after the first adsorption experiment, the felt could be redispersed in de-ionized water under stirring for ten minutes. Then the metal ion concentration in this water could be analysed and the felt used again in a second adsorption test using the identical metal ion solution as in the first round. This would indicate of the felt can be recovered for several adsorption cycles and it simultaneously indicates if the metal ions bind very strongly to the lignocellulosic surface (chemisorption or complex binding) or if the metal ion adsorption is reversible (physisorption). I believe that the article would benefit a lot from these findings, for instance examining this for Cu-ions, especially in regard to the practical application of these felts.
  6. The graphs for flax miss the elements, as they are shown in the graphs for hemp. Please add the elements to their according energy peak for the flax gaphs.
  7. Please use a different term for “abatement”, e.g., “reduction”. This is usually used in legal affairs but not in scientific context.

I would kindly ask the authors to answer to the raised questions before I would reconsider the article for its publication in “molecules”.

Reviewer 3 Report

This work is focused on using hemp- and flax-based materials in the form of felts as biosorbents for the removal of metals present in aqueous solutions. Biosorption of Al, Cd, Co, Cu, Mn, Ni and Zn from a single solution by the two lignocellulosic-based felts was examined using a batch mode. The influence of various parameters (initial metal concentration, adsorbent dosage, contact time and pH) on efficiency of sorption were studied.

Although the topic of the manuscript is very current, the scientific quality of the manuscript is questionable. I miss the scientific interpretation of the experimental results.

I recommend supplementing the manuscript with a study of the sorption kinetics and analyse of experimental data by the Langmuir and Freundlich models. I recommend major revision of the manuscript.

Round 2

Reviewer 1 Report

The reviewer agrees with the revision for this manuscript, and suggests that the revised manuscript can be accepted for publication in Molecules.

Author Response

Thank you for comment.

Reviewer 2 Report

Dear authors,

Thank you very much for your reply and corrections. I only have one minor comment for the moment. Figure 3 is not presented correct (compare with Figure 5), the "reduction" has to start at "0%" and should increase.

Author Response

Thank for your comment.

However, this figure is correct.

Figures 3 and 5 are not comparable. In figure 3, there is no point at "0"; the first one being at 1 mg/L: this is the lowest concentration we have studied (highest reduction). In this graph, as the concentration increases, the reduction decreases as expected. Moreover, these curves were drawn with a contact time of 60 min. On the contrary, figure 5 shows the reduction as a function of time, and thus the more the contact time increases, the more the reduction increases. The first point at "0" is correct.

Reviewer 3 Report

The authors respected the opponents' comments and improved the quality of the manuscript. I recommend publishing the article in the revised form.

Author Response

Thank for your comment.